# Interaction between Mesenchymal Stem Cells and the Immune System in Rheumatoid Arthritis

**DOI:** 10.3390/ph15080941

**Published:** 2022-07-28

**Authors:** Darina Bačenková, Marianna Trebuňová, Radoslav Morochovič, Erik Dosedla, Alena Findrik Balogová, Petra Gašparová, Jozef Živčák

**Affiliations:** 1Department of Biomedical Engineering and Measurement, Faculty of Mechanical Engineering, Technical University of Košice, Letná 9, 042 00 Košice, Slovakia; marianna.trebunova@tuke.sk (M.T.); alena.findrik.balogova@tuke.sk (A.F.B.); jozef.zivcak@tuke.sk (J.Ž.); 2Department of Trauma Surgery, Faculty of Medicine, Pavol Jozef Šafárik University in Košice, University Hospital of Louis Pasteur, Rastislavova 43, 041 90 Košice, Slovakia; radoslav.morochovic@upjs.sk; 3Department of Obstetrics and Gynecology, Pavol Jozef Šafárik University in Košice, Hospital AGEL Košice-Šaca, 040 15 Košice-Šaca, Slovakia; erik.dosedla@nke.agel.sk (E.D.); petra.gasparova01@nke.agel.sk (P.G.)

**Keywords:** mesenchymal stem cells, rheumatoid arthritis, immunomodulation, fibroblast-like synoviocytes, T regulatory cells, anti-citrulline protein antibodies

## Abstract

Rheumatoid arthritis (RA) is an autoimmune disease that causes damage to joints. This review focuses on the possibility of influencing the disease through immunomodulation by mesenchymal stem cells (MSCs). There is an occurrence of rheumatoid factor and RA-specific autoantibodies to citrullinated proteins in most patients. Citrulline proteins have been identified in the joints of RA patients, and are considered to be the most suitable candidates for the stimulation of anti-citrulline protein antibodies production. Fibroblast-like proliferating active synoviocytes actively promote inflammation and destruction in the RA joint, in association with pro-inflammatory cells. The inflammatory process may be suppressed by MSCs, which are a population of adherent cells with the following characteristic phenotype: CD105+, CD73+, CD90+, CD45−, CD34− and HLA DR−. Following the stimulation process, MSCs are capable of immunomodulatory action through the release of bioactive molecules, as well as direct contact with the cells of the immune system. Furthermore, MSCs show the ability to suppress natural killer cell activation and dendritic cells maturation, inhibit T cell proliferation and function, and induce T regulatory cell formation. MSCs produce factors that suppress inflammatory processes, such as PGE2, TGF-β, HLA-G5, IDO, and IL-10. These properties suggest that MSCs may affect and suppress the excessive inflammation that occurs in RA. The effect of MSCs on rheumatoid arthritis has been proven to be a suitable alternative treatment thanks to successful experiments and clinical studies.

## 1. Introduction 

Rheumatoid arthritis (RA) is a chronic systemic disease that causes damage to joints, connective tissue, muscles and tendons [1]. Chronic inflammation is a hallmark of RA joint pathology synovitis, which causes cartilage and bone tissue erosion by resident synoviocyte-like fibroblasts (FLS). Joint inflammation is initiated and maintained by autoimmune mechanisms [2]. The preclinical phase of RA, which takes a long period of time before the first clinical symptoms appear, is characterized by the presence of circulating autoantibodies, and increased levels of inflammatory cytokines and chemokines. In RA, inner lining hyperplasia occurs and the numbers of macrophages and fibroblast-like synoviocytes (FLSs) are elevated (Table 1). Inflammatory cells penetrate to the rheumatoid joint, where they are activated and contribute to local destruction. More specifically, they are mainly neutrophils that accumulate in the synovial fluid, absorb immune complexes, and release proteolytic enzymes [3].

Inflammation in the synovium is accompanied by the formation of a pannus in the joint lining. A pannus is an aggressive front tissue that destroys the local joint structures. Panus synoviocytes produce microscopic structures called lysosomes that release enzymes (proteins) called matrix metalloproteinases (MMPs), which have a degrading effect on cartilage. Altered cell metabolism ultimately leads to joint erosion and damage. The exact cause of RA is currently not fully understood. It is most likely to be a disease that occurs in genetically predisposed individuals and under certain circumstances, for example, as an acute disease in the presence of pathogenic microorganisms. RA disease is characterized by chronic inflammation, which is maintained by autoimmune processes. The course of RA is highly variable. Overall, however, the course is progressive and often leads to disability. The goal of treatment is to achieve remission, but the treatment must be timely and aggressive [1,2,3]. RA treatment is aimed slowing the progression of the disease by inhibiting the inflammation of the affected parts. Currently applied treatment methods include synthetic and disease-modifying anti-rheumatic drugs (DMARDs), non-steroidal anti-inflammatory drugs (NSAIDs) and glucocorticoids. Several side effects are present in patients following the glucocorticoids’ administration. A higher dose of glucocorticoids causes parchment skin, leg edema, sleep disorders, immunosuppression, weight gain, hypertension and diabetes. Recently, antirheumatic drugs of the type DMARDs, and especially the DMARD methotrexate (MTX), have been used therapeutically; furthermore, new biological substances have improved the treatment of RA. Interestingly, early treatment more effectively increases its effectiveness and reduces the extent of joint damage. However, there is a certain problem with catching and treating patients in the early stages of the disease; NSAIDs are indicated as therapy in combination with other medications during the RA phase. This is mainly a symptomatic treatment. Undesirable side effects are common with this therapy. DMARDs objectively suppress the inflammatory reaction, including the inhibition of the immediate phase reaction, and also reduce damage to the joint, which has been confirmed by X-ray analysis. The most frequently used drugs are sulfasalazine and methotrexate. Biological therapy is a modern alternative to RA therapy. In principle, it involves blocking the activity of the main inflammatory cytokine, tumor necrosis factor alpha (TNF-α). There are several preparations based on the neutralizing antibodies against TNF-α. Biological therapy is recommended for patients with a severe form of RA, mainly for patients who respond poorly to DMARDs treatment. Undesirable effects may occur, such as infections, tuberculosis, and malignant diseases. Biologic DMARDs and NSAIDs, which are known as effective painkillers, as well as other expensive drugs with a high potential to reduce disease symptoms and prevent disease progression in patients, are most commonly used in RA therapy. Cytokines such as TNF-α, Interleukin 1 (IL-1), IL-6, IL-7, IL-15, IL-17, IL-18, IL-21, IL-23, IL-32, IL-33 and granulocyte-macrophage colony stimulating factor (GM-CSF) are important in the development of RA. Interestingly, the results of clinical studies that focused on blocking IL-1, IL-17 and IL-18 did not show significant success. In contrast, blocking TNF-α or IL-6 was successful in influencing symptoms [4]. The negatives of RA treatment are its serious side effects. Moreover, treatment is not effective for all patients. In RA, up to 30% of patients do not respond to conventional treatment [5].

### 1.1. The Immunology of Rheumatoid Arthritis

Autoimmune diseases are manifested as disturbances in the inflammatory process, while the affected area is disturbed and its functionality is lost in the case of neglecting the necessary therapy. Changes in the joint during the RA process are related to an unbalanced biochemical process of catabolism and anabolism, while structural disorders occur in the joint. These serious biochemical changes that occur in connection with inflammatory synovitis in RA have the effect of suppressing the natural remodeling of structures in the joint. RA disease is characterized by the presence of autoantibodies in the period before the manifestation of the disease itself [6]. The occurrence of anticitrulline antibodies is related to the formation of immune complexes in the affected joint. The course of the autoimmune disease RA involves changes in both innate and adaptive immunity in subsets of T-lymphocytes, the cells of the monocyte, and the macrophage lineage. Among the manifestations of RA is a decrease in immune tolerance, which includes decreased T-reg cells. On the contrary, inflammatory cells and their activation increase, which is also related to the increased production of cytokines, which ends in chronic inflammation. The pro-inflammatory cytokines IL-6 and TNF-α are present at the site of inflammation. Due to the extent of the disease and the multiple affected parts, the disease is perceived as a systemic disease. Synovial hyperplasia is a very common manifestation of RA after joint injuries, and the stroma of the synovial tissue contain cells with a characteristic MSC phenotype. According to the available data, it has been demonstrated that MSCs can migrate from the BM into the joint cavity, and they have been detected in the proliferating synovial tissue. Authors de Bari et al. demonstrated the presence of MSCs in the human synovial membrane which have a characteristic phenotype and differentiation ability. MSCs, as stem cells, have an important role in homeostasis, while in the inflammatory process this balance is dysregulated [4,5].

### 1.2. Autoantibodies in Rheumatoid Arthritis

Monitoring the presence of autoantibodies in patients, rheumatoid factor (RF) and the autoantibodies to post-translationally modified proteins such as citrullination (anticitrulin antibodies) are now increasingly important in the diagnosis of disease. Autoantibodies develop in the serum and synovial fluid (SF) of patients with RA. Antibodies cause the formation of immune complexes in the joints, which results in the activation of immune cells through the complement system, as well as the direct activation of immune cells and the secretion of chemokines and cytokines that enhance the immune response. RF antibodies that recognize the Fc-terminus of immunoglobulins (Ig)-Gs are the standard and first type of autoantibodies found in RA. Now, it is known that the RF response uses a wide-range spectrum of isotypes, including IgM, IgG and IgA. The RF antibody is present in about 75% of RA patients, and has also been determined in other diseases and non-rheumatic conditions such as leprosy, syphilis, pulmonary tuberculosis, chronic liver disease, and sarcoidosis, as well as in many rheumatological diseases such as systemic lupus erythematosus (SLE) and Sjogren’s disease. In the current infection, RF can act to bind IgG-coated pathogens and condition their removal in the form of immune complexes [6]. Genetic factors have been shown to account for about 60% of the risk of developing RA. Anti-citrulline protein antibodies (ACPA) and carbamylation (anti-CarP antibodies) are very important. A risk factor for ACPA-positive RA is the HLA class II molecule HLA-DRB1-“shared epitope” (SE) alleles. Recent studies suggest that the HLA-DRB1 alleles, which encode SE, a five-amino acid sequence motif at residues 70–74 of the HLA-DRβ chain, are significantly present in severe RA. Another interesting assumption is the refinement of the SE motif, which relates to position 70 of the DRβ chain. Current facts suggest that while glutamine or arginine at position 70 appears to be a significant risk for RA, aspartic acid at this position provides protection [7]. The risk is much higher for people carrying one SE or two SE alleles compared to SE-negative individuals [8]. The incidence of RA by gender is higher in women compared to men, and the incidence rate in women is more than double. Due to a higher prevalence in women, reproductive and hormonal factors are thought to affect the disease. The exact role of gonadotropins, estrogens and androgens in RA is currently unknown. Estrogens are thought to be more pro-inflammatory and androgens more anti-inflammatory. The results of the studies are not yet clear [9]. There is a form of seronegative RA that is characterized by the absence of the autoantibodies RF and ACPA in serum. In addition, there are several differences in etiopathogenesis and different risk factors. Several authors have summarized data from studies that describe a non-seropositive form of RA. The RA treatment regimen does not currently take into account the presence or absence of autoantibodies in patients, and the therapy is largely identical [10].

### 1.3. Post-Translational Modifications of Proteins ECM

Specific serological markers of RA are antibodies to citrullinated proteins, which show higher specificity in RA compared to RF [11,12]. ACPAs specifically bind to citrullination after translational modification (PTM). ACPAs are synthesized by immune cells in the panus; thus, the trigger for ACPA production is the autoantigen present in the affected joint. Citrullination has been found in several body parts, as well as in both mouse and human inflamed joints. It is a PTM that can alter the interaction properties of proteins that contain lysine and arginine in their interactive domains. As previously mentioned, the post-translational process of citrullination, which is driven by calcium and the enzyme peptidyl arginine deiminase (PAD), leads to the removal of a positive charge for each arginine residue that is converted to neutral citrulline [13]. Intracellular enzyme PAD is an evolutionarily conserved protein with several isoforms in both mice and humans (PAD1-4 and PAD6). While PAD4 is present in monocytes and macrophages, both PAD2 and PAD4 are characteristic of SF. Both apoptosis and necrosis are associated with a locally increased calcium concentration, which also releases PAD. It is thought that even in inflamed tissues, PAD could act on the citrullination of ECM proteins such as fibrinogen and collagen. Citrulline collagen II, α-enolase, fibrinogen and vimentin have been detected in the joints of patients with RA, and are considered to be the most suitable candidates for the stimulation of ACPA production [14]. In donors who provided serum years before their first RA symptoms appeared, the ACPA test sensitivity was about 50%. Interestingly, ACPA autoantibody serums were detectable in several patients aged 10–20 years before the onset of the disease [15]. Post-translational changes in fibronectin and collagen proteins that are part of the extracellular matrix (ECM) can potentially affect cell properties such as adhesion, physicochemical properties and antigenicity [16,17]. One of the multifactorial causes of RA is associated with dysregulated ECM remodeling, which is an interesting fact for citrullinated vimentin. Citrullination, which has a negative effect on the assembly of vimentin filaments, leads to the increased formation of soluble precursors that are transported extracellularly. The extracellular localization of these precursors subsequently induces autoimmune responses in joints with RA. Citrullinated vimentin expression increases following the cell damage. It is also strongly expressed in the repair modulating cells, which stimulates their migration. The local production of posttranslational modified vimentin in the synovium of RA patients is expected. This potential mechanism of posttranslational modified vimentin (PTMV) formation could be related to inflammation. Macrophages and neutrophils that migrate from the periphery have high levels of PAD-2 and PAD-4 [18]. Vimentin is most commonly localized in the cytoplasm and nucleus, and to a lesser extent on the cell surface in the form of post-translationally modified phosphorylated vimentin. The endoplasmic reticulum and Golgi complex are involved in the process of vimentin formation, forming vesicles that are secreted extracellularly as phosphorylated vimentin. The cytokines present may regulate the expression of phosphorylated vimentin on the cell surface. Interestingly, MSCs produced by TNF-α and IL-10 reduce vimentin protein exocytosis. The exact physiological function of citrullination is not fully understood; however, it is known to cause the increased susceptibility of proteins to proteolytic degradation. Another interesting finding is that these proteins are processed antigen-present cells (APCs), which can be citrullinated prior to their presentation to T cells [19,20]. It is also interesting to point out that there is an involvement of citrullinated proteins in the pathogenic mechanism of RA in animal models. Hill et al. described the induction of RA in a transgenic mouse with the HLA-DR4-IE gene by citrulline fibrin, which is commonly found in inflamed synovial tissue, and is also a common target of autoantibodies in RA patients [21]. RA disease in antigen-induced mice was characterized by synovial hyperplasia. Furthermore, citrullination has been found to induce immunological intolerance for endogenous reasons. Citrullinated proteins significantly induced synovial arthritis in animals. Higher numbers of CD4+ T cell were recorded in the synovium of RA animals. T cells altered by antigen-presenting cells expressing citrullinated peptides appear to be a putative cause of T cell-dependent ACPA production and joint damage [22] (Figure 1).

## 2. Mesenchymal Stem Cells

Progress in cell therapy research has now been made, and the possibilities of regenerative procedures in patients with autoimmune diseases are being intensively investigated. Mesenchymal stem cell (MSC)-based therapy is a new alternative to the standard treatment of autoimmune diseases. It has been shown that several features of MSCs are suitable for the treatment. In the last century, MSCs were identified to be part of the bone marrow stroma as heterogeneous cell populations. They are capable of self-renewal and tissue regeneration, and have strong immunosuppressive properties [23]. MSCs are relatively easy to isolate from different parts of the organism, and are therefore ideal for in vivo and in vitro testing. MSCs have been described by Friedenstein as adherent, fibroblast-like cells with high self-renewal capacity [24]. Another characteristic feature of MSCs is their multilineage differentiation ability in vitro and in vivo. In particular, differentiation into multiple cell types, osteoblasts, chondrocytes, adipocytes, myocytes, epithelial cells and cardiomyocytes has been described [25,26,27,28,29,30,31]. Due to the absence of a unique marker, the MSC phenotype is defined by the expression of markers CD105, CD73, and CD90, and at the same time by the absence of the expression of the hematopoietic markers CD45, CD34, CD14, CD11b, CD79a, and CD19, and the MHC II molecule class [32] (Table 2).

Friendenstein was the first to describe the growth of so-called “colony-forming units—fibroblasts” (CFU-F) in bone marrow cultivation. The ability to form colonies still remains an important test for the quality of the examined cell sample. CFU-F is an indicator of the in vitro sample quality, recording the frequency of MSCs [33]. Bone marrow is the traditional source of MSCs. Another preferred source of MSCs is adipose tissue [34]. The isolation of MSCs is possible from many parts of the body’s organs and tissues. A promising source of MSCs is the chorion, part of the amniotic sac and the placenta [35,36,37]. The placenta is a rich source that contains two types of MSCs: amniotic mesenchymal stromal cells (AMSCs) and chorionic mesenchymal stromal cells (CMSCs). CMSCs can be isolated from the chorionic mesoderm layer [38,39,40]. CMSCs express the embryonic markers octamer-binding protein 4 (OCT4), SRY-related HMG-box 2 (Sox-2), homeoprotein Nanog, stage-specific embryonic antigen-4 (SSEA-4), and GATA binding protein 4 (GATA-4) [41] (Table 3). Overall, the main criteria for defining MSCs are as follows: cell adhesion to plastic, a specific surface antigen expression pattern, and specific differentiation potential towards osteogenic, adipogenic and chondrogenic lineages. The ability of MSCs to differentiate into cartilage, which is present in joint tissues, promotes therapeutic interventions by replacing damaged cells [42].

### 2.1. Mesenchymal Stem Cells in Synovium, in Their Native Environment

Synovium is an important tissue in RA, and its membrane lines the synovial joint cavity. Synovium has a lubricating effect on articular surfaces and provides nutrition to articular cartilage. Cells that exhibit the MSC phenotype can be isolated from synovium, where they are located mainly in the lining niche and the sublining perivascular niche. They may function as a reservoir of stem cells to repair joint structures, such as articular cartilage, with a very limited repair potential. In addition to MSCs, FLSs are an important part of the synovial membrane. In vitro, it is possible to isolate a mononuclear fraction of cells from a synovial membrane that contains both FLS and MSC. It is currently thought that MSCs may be involved in the anti-inflammatory processes of RA. In certain circumstances, the pro-inflammatory effects of MSCs are not ruled out. In an inflammatory environment, the MSC populations may be affected by inflammatory cytokines. It may be assumed that MSCs might have similar—but not exactly the same—functions. Synovial MSCs have been described as phenotypically heterogeneous, with similar populations of MSCs. An interesting fact is that the number of MSCs with in vitro multiline potential was significantly lower in the SF of RA patients compared to patients with osteoarthritis [43].

The effects of the cytokines present in the RA process on cell populations were observed experimentally. TNF-α prevents the differentiation ability of MSCs on the chondogenic line in vitro, and thus may contribute to the reduced ability of MSCs to repair joints and regenerate cartilage and bone during RA. Interferon gamma (IFN-γ) acts on the MSC as an activator of immunosuppressive and pro-inflammatory properties at the site of inflammation. Inflammation that occurs after tissue damage is usually accompanied by the infiltration of macrophages and neutrophils. This process is associated with phagocytosis by a macrophage, and is also accompanied by the release of pro-inflammatory factors IFN-γ, TNF-α, and IL-1. Furthermore, the body’s response to damage affects adaptive immunity and causes the activation of CD4+ and CD8+ T cells and B cells. The immunomodulatory role of MSC pro-inflammatory cytokines—IFN-γ, TNF-α and IL-1—can stimulate the immunosuppressive abilities of MSCs in a process that has been named “licensing”. Licensed MSCs with increased immunomodulatory capacity produce anti-inflammatory cytokines [6]. In an in vitro culture of FLS and MSCs, the transcription factor nuclear factor-κB is present. It acts to inhibit the differentiation capacity of MSCs, and at the same time stimulates ECM degradation processes [44]. Inflammatory processes in the joint affect the phenotype of cell populations, and an effect on the phenotype of FLSs and MSCs cells is expected. These populations may support further disease progression. It can be assumed that FLSs can also circulate and attack unaffected joints [45].

### 2.2. MSCs and Immune Cells in the Inflammatory Environment of Damaged Tissue

Chondroprogenitors have been described in areas of cartilage damage due to the inflammatory process. These could also originate from differentiated MSCs from the bone marrow. It can be assumed that MSCs migrate locally in the joint, and that reparative processes are limited, which mainly concerns the cartilage. MSCs with proven multilineage mesodermal potential can influence the disturbed imbalanced synovial environment. High levels of inflammatory cytokines in RA can also have a suppressive effect on stem cells. TNF-α suppresses the multilineage differentiation capacity of MSCs in vitro. TNF-α has a catabolic effect on cartilage and bone tissue. In patients with RA, the incidence of MSCs in synovial fluid is lower compared to MSCs in a healthy control group. MSCs have been found to have some pro-inflammatory properties in chronic inflammatory environments [14]. Inflammation that occurs after tissue damage is usually accompanied by the infiltration of macrophages and neutrophils. This process is associated with phagocytosis by a macrophage, and is also accompanied by the release of pro-inflammatory factors. Furthermore, the body’s response to damage affects adaptive immunity and causes the activation of CD4+ and CD8+ T cells and B cells. MSCs secrete several cytokines that have a direct effect on damaged tissue and a positive paracrine effect on the repair. MSCs are considered stem cells that cooperate in the repair of damage and are able to migrate to the site of injury. At the site of the defect, they cooperate with several types of inflammatory cells. According to the available data, MSCs are capable of secreting many cytokines and growth factors. Several of the mentioned factors are generated upon the principle activation of factor-κB (NF-κB) after exposure to pro-inflammatory stimuli such as IFN-γ, TNF-α, IL-1β, and lipopolysaccharide (LPS). These are TGF-β, FGF, vascular endothelial growth factor (vEGF), platelet-derived growth factor (PDGF), insulin-like growth factor 1 (IGF-1), and stromal cell-derived factor 1 (SDF-1) [23].

#### 2.2.1. Homing and Migration of MSCs

An environment with a higher level of cytokines stimulates the differentiation of progenitor cells into fibroblasts and endothelial cells, which directly participate in tissue repair. In addition, in MSCs there is an adaptation in the expression of molecules such as intracellular ICAM-1, VCAM-1, and galectins. MSC-mediated therapy is conditioned by the ability of MSCs to home in on and migrate to the site of the injury. MSCs have the ability to attach to endothelial cells or ECM proteins such as collagen, and fibronectin and laminin thanks to adhesive molecules and integrins. According to the available data, MSCs are able to transmigrate through the endothelium and basement membrane into tissues [29]. It is interesting that the adhesion molecules CD106 (VCAM-1), CD54 (ICAM-1), and ICAM-2, which are found on endothelial cells, are also expressed by MSCs. Circulating MSCs are able to effectively travel to damaged sites through the adhesive molecules on their surface: CD24, CD29 (β1-integrin) and CD44. MSCs express the CD44 transmembrane glycoprotein, which can act as a ligand that mediates the adhesion of hyaluronic acid. The adhesion molecules late antigen-4 (VLA-4) and P-selectin are involved in MSC transmembrane migration. In addition, MSCs are characterized by the expression of multiple integrin receptors—α1, α2, α3, α4, α5, β1, β3 and β4—associated with cell–cell contacts and adhesion to extracellular matrix proteins. It is important to note the strong interaction of VLA-4/VCAM-1 particles, which are critical receptors for MSC transendothelial migration [31].

#### 2.2.2. The Modulation of T Cells by MSCs

T-lymphocytes are among the main immune cells that influence inflammation in RA. Resident joint lymphocytes and MSCs that are present in the joint show mutual interactions. T lymphocytes directly affect MSCs, mainly through the effect of cytokines, IFN-γ and TNF-α, which stimulate the migration of MSCs, and at the same time block the differentiation capacity. MSCs have an inhibitory effect on the proliferation of T helper (Th) and cytotoxic T lymphocytes. In addition to the influence of cytokines, the subpopulation of Th1 and Th17 cells plays a significant role in degradation processes. IL-17-related cytokineTh17 suppresses the chondrogenic differentiation of MSCs by suppressing the chondrogenic transcription factor SRY-Box TF9 (SOX9) and its activator protein kinase A (PKA). The relationship between the influence of T cells and MSCs is reciprocal, and MSCs also influence T cells. MSCs have a stimulatory effect on the differentiation of Th2 and regulatory T cells (Treg), leading to an anti-inflammatory effect. In vitro culture MSCs have a suppressive effect on the proliferation of T lymphocytes in healthy individuals. MSCs have the effect of increasing the percentage of T-reg and act to maintain a tolerogenic immune state. Several conflicting results of the in vitro testing of RA patients’ MSCs to suppress T-lymphocyte proliferation suggest that the outcome could probably correlate with the disease severity [29].

#### 2.2.3. Interactions between MSCs and Dendritic Cells

MSC have natural inhibitory effects on dendritic cells (DCs). They have a suppressive effect on the differentiation of monocytes into DCs. They also dampen the switch from monocytes to macrophages. This is related to the ability of MSCs to inhibit DC maturation and antigen presentation. MSCs expressing IL-1Rα can inhibit IL-1 production by DCs in co-cultures. It is a very interesting finding that RA DCs have a high expression of the pro-inflammatory transcription factor NF-κB, which is directly related to the production of inflammatory mediator TNF-α. It has been shown that monocyte-derived DCs can acquire a suppressor cell phenotype under the influence of MSCs. DC-modulated MSCs are capable of IL-10 and IL-4 secretion and the reduction of IL-12 and IFN-γ. DCs in the setting of RA indicate a failure of immune control mechanisms that could affect the activity of MSCs. MSCs could be involved in RA, and could act on DCs through TLR activation. TLR2 and TLR4 are activated in RA in the synovial environment. The cytokines IL-12 and IL-18 and IFN-γ act to increase TLR4. As a result, MSCs can produce pro-inflammatory IL-6. TLRs related to the pro-inflammatory functions of MSCs are evident in RA. MSCs and DCs behave differently in autoimmune diseases through pro-inflammatory cytokines [5,43].

#### 2.2.4. Macrophages and MSCs

MSCs also act on monocytes, which differentiate through their influence on the anti-inflammatory alternative M2 phenotype. The influence of IDO, TGF-β, IL-10, and PGE-2 is very important in this process of immunomodulation. This process is mainly influenced by PGE2 and IDO, which strengthen the anti-inflammatory function of macrophages. M2 macrophages secrete IL-10, and this also has an effect on the frequency of Treg cells, which affects the reduction of tissue neutrophil migration. MSCs have a suppressive effect on the proliferation and cytotoxic activity of NK cells. This further affects B cells and antibody secretion [43]. Macrophages can produce a bone morphogenetic protein (BMP) which stimulates the proliferation and osteogenic differentiation of MSCs. Resident macrophages in inflamed joints in RA are acted upon by IL-1 and TNF-α, which stimulate Th1, acquire a pro-inflammatory phenotype, and cause a hyperimmune response. Pro-inflammatory M2 macrophages can stimulate RANKL expression by MSCs to support the formation of osteoclasts. They secrete matrix metalloproteinases (MMPs), which cause cartilage ECM degradation, and TNF-α promotes the formation of osteoclasts in an inflammatory environment. TNF-α stimulates the apoptosis of chondrocytes; on the contrary, it also stimulates the proliferation of osteoclasts and increases the degradation of the cartilage matrix. Cartilage repair is weakened in the RA environment. IFN-γ is a modulatory factor that acts as a signal to activate T cells. However, in the presence of MSCs, it suppresses the proliferation of T cells while activating MSCs and acting to increase their immunomodulating functions. The mechanism of immunosuppression reduction by MSCs is not fully understood. The overall interactions of MSCs and immune cells appear to be very complicated and multilevel. The properties of MSCs in a chronic inflammatory environment appear to be limited [43].

#### 2.2.5. B Cells and MSCs

Recently, RA disease was considered to be mediated mainly by T lymphocytes and macrophages. Currently, the potential of anti-CD20 therapy in RA has been reported. It was found that the depletion of memory B-cell subpopulations in patients caused the remission of RA protection. B cells are phylogenetically the youngest, and are mainly precursors of antibody-secreting cells. The dysregulation of B cell responses can potentially cause autoimmune disease. MSCs are known to exert an inhibitory effect on the proliferation and function of B cells. In contrast, the FLS cell population could mediate B-lymphocyte migration via SDF-1 and VCAM-1. There is a similar connection between MSCs, B cell populations with VCAm-1 expression in MSCs, and VLA-4 expression in B cells. RA synovial fibroblasts could promote B cells’ survival. The function of MSCs against B-lymphocytes in RA is currently being studied intensively [45].

### 2.3. Fibroblast-like Synoviocytes’ Interactions with Immune Cells

Synovial fibroblasts have been defined as non-vascular, non-epithelial cells in the synovium that arise by dividing locally during embryogenesis, with the ability to produce fibrous matrix proteins. It is generally accepted that MSCs and synovial fibroblasts are different functional stages derived from the stromal cell lineage, and these spindle cells are morphologically similar. RA is characterized by an increased proliferative potential of FLS involved in the panus formation. FLS non-immune synovial tissue cells act by destroying joints via various mechanisms through the interplay between infiltrating inflammatory immune cells. In the RA process, FLSs are activated by the endogenous ligands of pattern recognition receptors (PRRs) and inflammatory cytokines. In this process, articular cartilage and bone are damaged. The synovium contains the lining layer and macrophage-type cell populations, FLSs, and the lining of free connective tissue scattered between the endothelium and small blood vessels with pericytes. Fibroblasts can be assumed to have functionally different properties depending on their location. Synovial FLS has similar characteristics consistent with the common fibroblast phenotype in the presence of type IV and V collagens, vimentin and CD90 (Thy-1), vascular cell adhesion molecule-1 (VCAM-1), and CD44 [5]. FLSs in the synovium express uridine diphosphoglucose dehydrogenase (UDPGD) for the synthesis of hyaluronan, an important component of SF. At the same time, they produce lubricin, a protein that allows joint lubrication. Furthermore, FLSs express the adhesion molecule cadherin-11, which causes the aggregation of fibroblats in vitro and in vivo. Cadherins are involved in cell adhesion through binding to cadherin of the same type on a neighboring cell by extracellular N-terminal first cadherin domains. They affect the stability of the ECM [5,44,46]. Isolated cells adhering to plastic from RA synovium, commonly considered FLS, have the ability to erode cartilage via MMPs, as in RA, and also via the presence of cells with a mesenchymal multipotency of MSCs. The relationship between MSCs and the FLS in the synovial lining layer is interesting and generally unclear [47]. According to the authors, MSCs could occur in the lining layer as stem cells together with FLS and macrophages [5,43].

Cellular interactions between FLS and hematopoietic immune cells may play an important role in RA, as well as in the production of autoantibodies in RA synovium. The localization of CD4+ T lymphocytes and B lymphocytes in hyperplastic RA synovium suggests their role in the pathogenesis of RA. CD4+ helper T lymphotytes are an integral part of the etiology of RA. CD4+ T lymphocytes differentiate helper T lymphocytes (Th17 cells) producing IL-17, and participate in RA. Th17 cells act to activate FLS, macrophages, endothelial cells and chondrocytes by IL-17 [48]. T cell suppression has been reported to be effective in RA therapy. FLSs are involved in T cell differentiation through cytokine production. Foxp3 + T-regulatory lymphocytes (Tregs) are formed in the thymus, but can also be formed in peripheral regions or induced in vitro by TGF-β. TGF-β affects the differentiation of several types of T cell subgroups, inducible T-reg (iTreg), Th17 and Tph cells, and the induction of FoxP3 transcription factors. RA FLSs are highly expressed by chemokine C-X3-C motif ligand (CX3CL1) (fractalkine); furthermore, its C-X3-C motif chemokine receptor 1 (CX3CR1) expression is upregulated in the CD4+ and CD8+ T cells of RA patients, suggesting a role for CX3CL1/CX3CR1 in the pathogenesis of RA. An important mechanism of action of polyclonal tTregs antibodies is the suppression of T-effector lymphocyte delivery to the target organ, while antigen-specific Tregs primarily prevent T-cell activation by affecting antigen-presenting DC [49,50,51].

### 2.4. Immunomodulatory Potential of Mesenchymal Stem Cells

As reported by several authors, the biological properties of MSCs are directly affected by the inflammatory environment. MSCs are generally quiescent, with a lower expression of molecules active in immunoregulation. MSCs affect the suppression of the inflammatory environment by modulating the function of immune cells. The action of MSCs on immune cell function is currently thought to be mediated by factor secretion and through cell–cell contact and extracellular vesicles [52,53,54,55,56] (Figure 2).

MSCs are able to switch on to modulate the immune system. While the immune system is not sufficiently activated, MSCs have the ability to promote inflammation. In contrast, MSCs are able to modulate and reduce inflammation when it is necessary to suppress the activity of immune cells and the production of pro-inflammatory cytokines [57]. MSCs express toll-like receptors (TLRs), specifically TLR2, TLR3, TLR4, TLR7 and TLR9. TLR expression is location and tissue dependent. TLR activation can act to stimulate immune cells as well as MSCs. In clinical studies and RA models, TLRs have been found to contribute to the pathogenesis of the disease. TLR3 and TLR4 expression in human synovium has been determined [58]. TLRs play an important role in switching the excellent features of MSCs. The MSC1 phenotype is mostly associated with the early phase of inflammation; it is activated by TLR2 or TLR4 receptors. The activation of TLR3 induces the anti-inflammatory phenotype of MSCs—which is also known as MSC2—until TLR4 activation induces a pro-inflammatory phenotype, MSC1. Pro-inflammatory MSC1 releases IL-6 and IL-8, and promotes macrophage polarization to the M1 phenotype. MSC1s are further capable of enhancing T cell responses by secreting chemokines: macrophage inflammatory proteins (MIP) MIP-1α and MIP-1β, “regulated on activation, normal T cell expressed and secreted” (RANTES), CCL5, ligand CXCL9, and CXCL10 [59]. This pro-inflammatory phenotype of MSCs is characterized by low levels of the immunosuppressive factors indolamine 2,3-dioxygenase (IDO) and nitric oxide (NO). In the absence of an inflammatory environment, at low levels of TNF-α and IFN-γ, MSCs may acquire a pro-inflammatory phenotype proliferation. The osteogenic differentiation of MCSs is increased at this stage, and chondrogenic and adipogenic differentiation is decreased. Influencing inflammation is important for physiological healing. However, in the case of chronic inflammation, it does not go into the healing phase, which creates an imbalance [60]. MSC1, especially in the early stage of inflammation, has neutrophil-mediated pro-inflammatory functions. When TNF-α and INF-γ reach elevated levels in the environment, MSCs are stimulated to secrete IDO, leading to the inhibition of T cell proliferation and the promotion of Tregs cells. Therefore, TNF-α and INF-γ stimulation has been proposed for the transition between pro-inflammatory and anti-inflammatory actions [56]. Thus, in the presence of an anti-inflammatory environment with high levels of TNF-α and IFN-γ, MSCs are activated and adopt an anti-inflammatory phenotype. They are able to migrate to the site of inflammation, into sites with a higher content of pro-inflammatory cytokines [61]. MSCs are able to modulate innate and adaptive immune responses, in particular by reducing the number of DCs, macrophages, natural killer cells (NK), and B and T cells, and by stimulating the anti-inflammatory phenotype T cells. Probably the most frequently analyzed immunomodulatory effects of MSCs are through secreted factors. Upon activation by TNF-α and INF-γ, MSCs act as paracrine cytokines by producing several soluble molecules, transforming growth factor-β1 (TGF-β-1), IDO, hepatocyte growth factor (HGF), prostaglandin E2 (PGE2), hemoxygenase (HO) and IL-10 [62] (Figure 3).

The mechanism of IDO action is influenced by the conversion of the amino acid tryptophan to kynurenine, which impairs the synthesis of several cellular proteins and suppresses T cell proliferation [63]. IDO also influences the formation of Tregs cells and MSC-induced tDCs [64]. An environment with elevated TGF-β levels via MSCs promotes Tregs cell production. MSCs can activate monocyte differentiation into M2 macrophages. MSCs can also inhibit the differentiation of monocytes into DC, and can direct them to a tolerogenic profile [65]. The MSC secretion of IL-10, TGF-β, PGE2, NO and IDO affects B cell proliferation, activation and maturation. It also affects immunoglobulin production, as well as C-X-C chemokine receptor 4 (CXCR4), CXCR5 and C-C chemokine receptor type 7 (CCR7) expression, thereby reducing B cell migration capacity. The in vitro results suggest that the intercellular contact of MSCs with CD3+ T cell populations is necessary to increase IL-10 levels. IL-10 is involved in Tregs cell production by MSCs, and increases programmed cell death protein 1 (PD-1) expression in CD4+ CD25+ cells, which is associated with immunomodulatory activity [52]. The expression of several other molecules is associated with the immunomodulatory function of MSCs, such as HLA-G1, PD ligand 1 (PD-L1), CD40 and Jagged-1 (JAG1). Activated MSCs further express adhesion molecules, VCAM-1/CD106, intracellular adhesion molecule 1 (ICAM-1/CD54) and CXCR4. Through adhesion molecules, cells have the ability to bind to the ECM, with the ECM molecules acting to migrate the MSCs and mediating their interaction with immune cells, thereby mediating the MSCs for immunosuppression. ICAM-1 and VCAM-1 are thought to have a significant effect on the MSC-mediated immunomodulation process, and are induced by interferon gamma and IL-1 [66]. At the position of inflammation, MSCs express CD90, an activated leukocyte adhesion molecule (ALCAM/CD166), and other integrins that mediate MSC interactions with T cells. Inflammation has been shown to affect MSCs by suppressing helper T cells 1 (Th1) and stimulating Th17 proliferation along with expansion of regulatory T cells (Tregs). According to recent work, MSCs are able to modulate the immune system by reducing the harmful Th1/Th17 response and enhancing the protective response of Tregs cells [67]. The inhibition of lymphocyte proliferation has been observed during the co-cultivation of MSCs and mitocin agglutinin-activated T cells [68]. In RA, the problem is the imbalance between the immune response and the ratio of pro-inflammatory and anti-inflammatory cells, especially between memory Th17 cells and memory Tregs cells, which is associated with the immunopathogenesis of RA. In summary, MSCs are thought to be able to affect the function of memory lymphocytes, including Th17 cells, in the promotion of Treg cell production [69,70]. Chemokines affect lymphocytes that move to sites of inflammation, and also bind to CCR5 and CXCR3 expressed on T cells. Activated MSCs have a greater capacity to act suppressively on NK cells at the site of inflammation.

TGF-β are cytokines with a pleiotropic role that are involved in different biological processes such as development, carcinogenesis, wound healing, hematopoiesi and immune responses. TGF-β isoforms have very similar responses in vitro; however, they show different functions in vivo [71,72]. TGF-β is produced by MSCs in the form of a homodimeric precursor associated with polypeptides consisting of the latency-linked peptide domain (LAP). Several enzymes can affect TGF-β activity. TGF-β and LAP form a latency complex (SLC). Latent TGF-β may be produced by the cell but may remain associated with the cell surface with several membrane-bound molecules. TGF-β activation is a regulated process, and it involves the dissociation of TGF-β from LAP, which allows TGF-β to bind to its receptors and induce signaling. This is achieved through several mechanisms, including the cleavage of LAP by either plasmin MMP −2 and −9, or interaction with integrins αVβ6 and αVβ8. Human MSCs produce TGF-β1 and also secrete TGF-β2 and TGF-β3 isoforms. MSCs have the ability to activate TGF-β via latent TGF-β binding protein-3 (LTPB-3) and thrombospondin −1 in vitro [73]. Interestingly, MSC production by TGF-β may be stimulated by several factors: elevated pro-inflammatory cytokines, TLR agonists, glucose levels, and hypoxia. Patients with inflammatory diseases have been reported to show altered TGF-β production compared to MSCs from controls [74]. In most cell types, TGF-β activates the canonical SMAD signaling pathway. MSCs express several TGF-β co-receptors, including CD105 (endoglin) [71].

### 2.5. Mesenchymal Stem Cells for RA Treatment

Regenerative medicine is one of the most promising areas of modern medicine in the 21st century. MSC-based RA treatments have recently become an interesting therapeutic option. MSCs have been shown to regulate a wide range of basic cellular functions [75,76]. Several clinical studies have recently been produced to evaluate the importance of RA MSC therapy [77]. The immunosuppressive properties of MSCs are currently being investigated for clinical application in a number of immune-mediated diseases. Allogeneic MSCs have not been shown to elicit an immune response in the host organism [78]. For the therapeutic use of multiplied MSCs, their precise phenotypic characterization is essential. After in vitro cultivation, their phenotypic stability is tested according to the criteria of the International Society for Cell Therapy [32]. Adult stem cells have found application in clinical practice, and MSCs are most often used in the repair of bone defects in orthopedics. Autologous MSC therapy has certain limitations and disadvantages. The main problem is attaining a sufficient therapeutic dose of a cell suspension that has the required quality, phenotype, and amount of MSCs. A serious problem is the collection of a sufficient amount of autologous MSCs from patients with autoimmune diseases. There are frequent problems with the age of patients, with increasing age decreasing the number of isolated clonogenic MSCs and decreasing the number of CFU-F colonies in an in vitro sample [33]. MSCs isolated from elderly individuals have reduced biological activity, differentiation ability, and regenerative potential. MSCs in systemic autoimmune diseases such as RA, systemic lupus erythematosus (SLE) and diabetes mellitus have altered properties. MSCs isolated from young and healthy donors are a promising solution; in this case, it is possible to isolate and multiply sufficient amounts of MSCs for therapy. Allogeneic MSCs can be stored and prepared quickly enough for therapeutic application. This fact is advantageous for the use of MSCs in the case of myocardial infarction. The limited expression of MHC class I, and the absence of expression of MHC class II and costimulatory molecules CD40, CD80 and CD86 are responsible for low immunogenicity. The non-immunogenicity of MSCs and their immunosuppressive properties are advantageous in MSCs’ allo-transplantation, and do not induce an immune response in the recipient [53].

The low number of clinical trials using MSCs is currently seen as one of the negatives of RA treatment, although the number of studies has increased approximately fourfold over the last 10 years. Nevertheless, the percentage of phase III or IV studies remains below 10% [79]. The use of allogeneic MSCs is being tested for RA therapy. In patients in the early stages of RA, the effective contribution of allogeneic MSCs in clinical trials is expected to restart the immune system by inducing its regulation. Therapy with autologous MSCs is possible; however, several limitations prevail. These facts affect the therapy and its outcome [80]. The authors of the study demonstrated the immunomodulatory properties of MSCs by suppressing CD19 + B cells in vitro, and were able to reduce ACPA production, which could have an effect in patients with RA. The authors described the mechanism of action of MSCs in the treatment of autoimmune diseases and the already mentioned induction of the de novo production of antigen-specific CD4 + CD25 + FoxP3 + Treg cells. Furthermore, through TGF-β production in the immune system, MSC suppresses Th17 effector cells and induces FoxP3 + Tregs, and thus maintains self-tolerance. The results of the study show that in vitro umbilical cord-derived MSCs (UC-MSCs) were able to inhibit FLS proliferation via IL-10, IDO and TGFβ1. The decrease in IL-6 FLSs secretion was also noted. UC-MSCs secreted PGE2 and TGF-β1. Furthermore, T cell-mediated activity was observed, and promoted the differentiation of CD4 + Foxp3 + Tregs cells from RA patients [76,81,82]. In another study, the authors determined the effect of human fat-derived MSCs on immune cells in RA patients. In vitro, MSCs were co-cultured with lymphocyte subsets of T, B and Treg cells [79]. The authors monitored the levels of the pro- and anti-inflammatory markers TNF-α, IFN-γ and IL-10 during co-cultivation. The authors attributed several recorded properties of MSCs through the mechanism of the therapeutic effect of MSCs in the study. They further studied the paracrine effect of MSCs, the production of anti-inflammatory and anti-apoptotic mediators, which is an immunomodulatory effect mainly in graft-versus-host disease (GvHD) monitoring, differentiating into cell types identical to the surrounding tissue [56,83]. In a recent study, the authors evaluated bone marrow MSCs administered intra-articularly to the knee in patients with RA. The patients were monitored for one year, and no side effects were reported after the administration of the MSCs. The results of the study show that an improvement—the prolongation of walking without pain—and pain reduction were reported according to the observed arthritis index of the Universities of Western Ontario and McMaster University. The exact effective dose of MSCs remains unclear, and depends on the therapeutic application. The authors used a single dose of approximately 40 million autologous MSCs in the study described [77,84]. Another study looked at the immunomodulatory effect of MSCs. The authors studied the effect of MSCs on the percentage of regulatory T cells. The study examined the safety and efficacy of allogeneic MSC transplantation in refractory RA. While the efficacy of the therapy was partial, the safety of allogeneic MSCs was demonstrated [85]. According to the latest findings, there is an effect of the release of extracellular vesicles (EVs) by MSCs, which have active immunoregulatory effects even at distant sites. EVs are classified mainly on the basis of their parameters into exosomes and microvesicles. Exosomes are homogeneous vesicles ranging in size from 40 to 100 nm that are derived from multivesicular bodies and secreted by the fusion of the multivesicular bodies with the cell membrane. Heterogeneous populations of microvesicles have a size in the range of 100 to 1000 nm, originating from the straight projections of the cell membrane, which separate from the surface. MSCs produce exosomes and MV with characteristic MSC markers (CD105, CD90 and CD73), and also PD-L1, Galalactins-1 (Gal-1) and TGF-β [52,86]. Exosomes mediate direct intracellular miRNA transfer between cells, and provide a new therapeutic option for RA. MicroRNAs (miRNAs or miRs) are smaller, non-coding RNAs that control metabolic processes among themselves and are highly associated with the pathogenesis of RA with high miRNA expression [87]. Pathological miRNAs in FLSs, osteoclasts and T cells, which affect the destruction of joints, condition the inflammation and degradation of the ECM and invasive behavior of cells. RA is associated with altered miRNA expression. In the described study of patients with RA and FLS, the expression of miR-150-5p was shown at a reduced level in RA, as in OA. On the contrary, the expression of vascular endothelial growth factor (VEGF) and MMP14 were increased in RA compared to osteoarthritis (OA). MiR-150-5p is involved in T cell maturation, and in the pathogenesis of RA. In addition to inner destruction, RA is also associated with neoangiogenesis. In this process, exosomes play a role in communication between cells, and miRNA is transferred. In the most recent studies, the effect of miR-150-5p exosomes was investigated on MMP14 and VEGF expression from RA patients on perinatal cell migration in the transwell system [88]. Several studies have shown that MSCs release paracrine factors that have anti-inflammatory effects. Experimental results confirm that MSCs inhibit human T cell proliferation and reduce the severity of collagen-induced arthritis in mouse models. As we have already mentioned, MSCs inhibit the activity of several populations of immune cells. Therapy with the application of allogeneic MSCs applied in a carrier from cancellous bone tissue, which were used in the repair of bone defects, had a comparable effect to autologous MSCs. MSCs with low immunogenicity have the ability to act in an immunomodulatory fashion in maintaining tolerance during transplantation, in the autoimmune response, and in tolerance between mother and fetus. MSCs can have an immunomodulatory effect on several processes in the body and related diseases, such as graft-versus-host disease (GvHD), RA, experimental autoimmune encephalomyelitis, and diabetes mellitus. MSCs have a high affinity for cancer tissue, and can thus help as auxiliary carriers of therapeutic agents [53].

## 3. Conclusions

Theadvanced development of cell therapies is a current topic, and cell therapy research is being investigated intensively in experimental animal models and in clinical studies. MSCs are a non-hematopoietic cell population with the secretory ability of bioactive molecules, as well as immunomodulatory and antifibrotic effects [54]. They are able to influence the activation of a large number of cells in the immune system. In addition, MSCs affect dendritic cell maturation by activating NK cells, and are able to inhibit lymphocyte proliferation and function, and stimulate Tregs [55,68]. MSCs inhibit the pro-inflammatory effect of neutrophil immune cell proliferation cytokines and the cytotoxic activity of resting NK cells. The immunosuppressive properties of MSCs are associated with hypoimmunogenicity and low levels of MHC class I molecules, as well as the absence of the expression of MHC class II and costimulatory molecules CD40, CD80, and CD86 [80]. In the etiology of RA, FLSs contribute to the pathogenesis of the disease. At present, the relationship/interaction between FLS and MSCs, in contrast to immune cells, has not been investigated sufficiently. In vivo and in vitro cooperation will be further investigated in detail. The facts suggest that FLS and MSCs participate in equilibrium in a healthy synovium, and act to inhibit T cell proliferation [5,89]. RFs, and in particular ACPAs, are thought to play a key role in autoimmune diseases such as RA. It is believed that the production of a modified protein antigen during inflammation is the most frequently presented molecular basis for the production of citrullinated RA proteins. These processes take place even after apoptosis, and initiate an immune response and the formation of autoantibodies against peptides or the whole protein [22]. ACPAs are associated with an HLA epitope, suggesting their specificity for RA [11]. In addition, negative factors have been reported to increase the expression of the PAD2 enzyme. Several authors have shown a correlation between smoking and the presence of autoantibodies to RA [6,90,91]. TNF-α and IFN-γ affect the activation and enhancement of the immunomodulatory properties of MSCs. The use of these cytokines at appropriate concentrations in vitro could promote the increased expression of immunoregulatory molecules. MSCs affected by the presence of pro-inflammatory cytokines respond by the secretion of IDO, NO and IL-10. These factors subsequently affect lymphocyte proliferation and function [62,68,92]. The use of extracellular vesicles, which are part of the secretome of MSCs, appears to be an interesting perspective for RA therapy. These small vesicles secreted by MSCs carry various biochemically active components in the lipid bilayer membranes, and have the ability to fuse with recipient cells. This allows direct communication between cells. MSC-derived EVs contain miRNAs with the ability to regulate transcription in terms of reducing inflammation. The anti-inflammatory properties of MSC-EV are effectively influenced by the action of TNF-α and IFN-γ. Furthermore, it is hypothesized that EVs derived from human MSCs could have a suppressive effect on RA synovial hyperplasia. In conclusion, it can be established that the use of MSCs in the treatment of RA is a promising option, but the optimization of the source of MSCs, the exact concentration, and the method of application of the therapeutic dose will be necessary [87]. The use of these MSC-activating cytokines at appropriate concentrations in vitro could promote the increased expression of immunoregulatory molecules. The full mechanism of action and the detailed scope of the therapeutic effect of MSCs has not been precisely described. Overall, MSCs, due to their immunomodulatory properties, contribute to the homeostasis of the body’s immune system.

## Figures and Tables

**Figure 1 pharmaceuticals-15-00941-f001:**
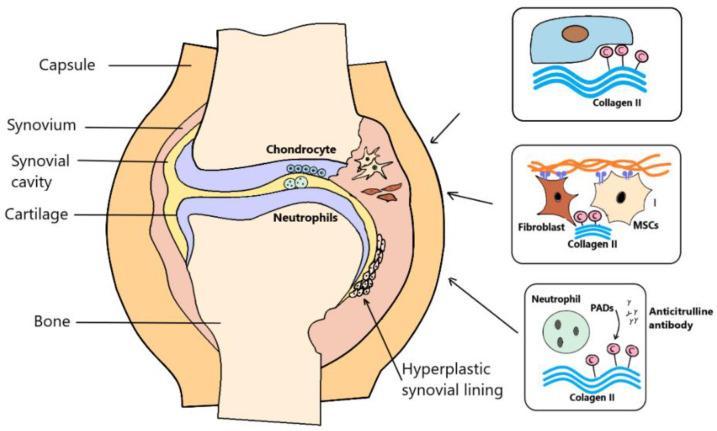
Antibodies to citrullinated proteins are specific serological markers of rheumatoid arthritis (RA). Anti-citrulline protein antibodies (ACPA) specifically bind to citrullination after translational modification. ACPAs are synthesized by immune cells in the panus, such that the trigger for ACPA production is the autoantigen present in the affected joint. Peptidyl arginine deiminase (PAD) are characteristic of synovial fluid. In inflamed tissues, PAD could act on the citrullination of extracellular matrix (ECM) proteins such as fibrinogen and collagen. Citrulline collagen II, fibrinogen and vimentin were found in the joints of RA patients.

**Figure 2 pharmaceuticals-15-00941-f002:**
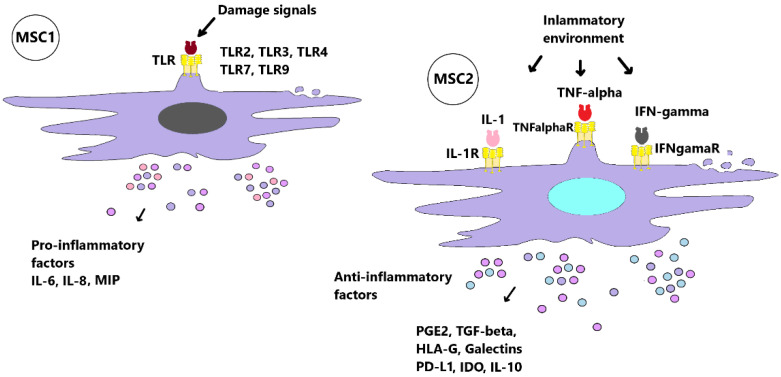
Pro-inflammatory mesenchymal stem cells 1 (MSC1) and the anti-inflammatory phenotype of MSC2. The schemes follow the same formatting that MSCs are able to switch on modulation of the immune system. MSC1s express Toll-like receptors (TLRs), specifically TLR2 and TLR4. The MSC1 phenotype is mostly associated with the early phase of inflammation, being activated by TLR2 or TLR4 receptors. Meanwhile, TLR4 activation induces a pro-inflammatory phenotype, MSC1; pro-inflammatory MSC1 releases interleukin 6 (IL-6) and IL-8, and promotes macrophage polarization to the M1 phenotype. MSC1 is further able to enhance T cell responses by secreting chemokines, i.e., macrophage inflammatory proteins (MIPs). In the presence of an anti-inflammatory environment with high levels of tumor necrosis factor alpha (TNF-α) and interferon gamma (IFN-γ), MSCs are activated and adopt an anti-inflammatory phenotype. The anti-inflammatory phenotype of MSC2 is characterized by high levels of the immunosuppressive factors indamine 2,3-dioxygenase (IDO), HLA-G, transforming growth factor beta (TGF-beta), galectins, and IL-10.

**Figure 3 pharmaceuticals-15-00941-f003:**
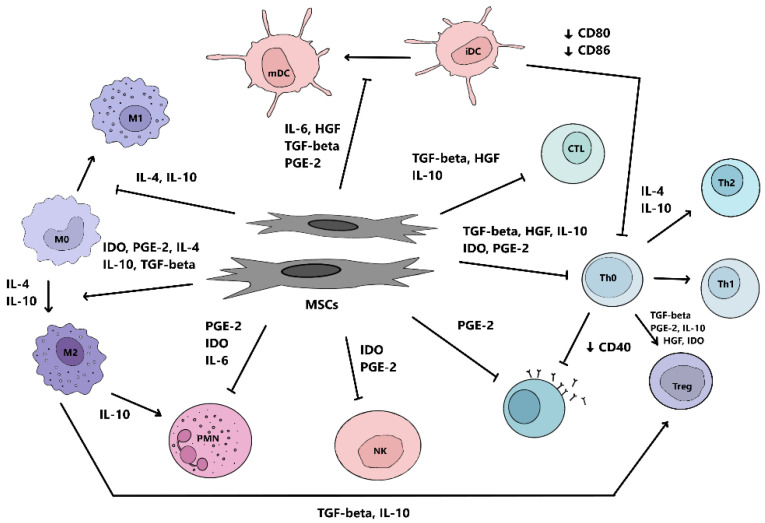
Interaction scheme between mesenchymal stem cells (MSCs) and the cells of the immune system. Activated MSCs secrete the following cytokines: prostaglandin E2 (PGE-2), indolamine 2,3-dioxygenase (IDO), nitric oxide (NO), and TGF-β. MSCs inhibit cytotoxic T cell proliferation and stimulate T helper (Th) cell production. The induction of Th2 differentiation and regulatory T cells (Treg) leads to the activation of an anti-inflammatory environment. Dendritic cell maturation (iDC) is inhibited by IL-6. At the same time, the expression of costimulatory molecules CD40, CD80 and CD86 acting to inhibit T cell activation is reduced. MSCs also act on monocytes, which target differentiation to the alternative anti-inflammatory phenotype M2, which is involved in the stimulation of Treg cells. MSCs are also able to inhibit NK cell proliferation, and to reduce their cytotoxic activity and cytokine secretion. They also suppress B cell proliferative activity, which leads to Ig secretion and production. Abbreviations: iDC, immature dendritic cell; IL, interleukin; HGF, hepatocyte growth factor; TGF-β, transforming growth factor-β; PGE-2, prostaglandin E2; IDO, indoleamine 2,3-dioxygenase; NO, nitric oxide; PD-L1, programmed death ligand 1; MSC, human mesenchymal stem cell; Treg, T regulatory; Th, T helper; CTL, cytotoxic T cell; mDC, mature dendritic cell; PD-1, programmed cell death protein 1; PMN, polymorphonuclear leukocyte; NK, NK cell.

**Table 1 pharmaceuticals-15-00941-t001:** Abbreviations often used in the text.

Abbreviations	Acronym
Anti-citrullinated protein antibodies	ACPAs
C-X-C chemokine receptor	CXCR
Dendritic cells	DC
Extracellular matrix	ECM
Extracellular vesicles	EVs
Fibroblast-like synoviocytes	FLSs
Immunoglobulins	Ig
Indolamine 2,3-dioxygenase	IDO
Interferon gamma	IFN-γ
Interleukin	IL
Intracellular adhesion molecule 1	ICAM1
Macrophage inflammatory proteins	MIP
Matrix metalloproteinases	MMP
Mesenchymal stem cells	MSCs
Natural killer cells	NK cells
Peptidyl arginine deiminase	PAD
Prostaglandin E2	PGE2
Rheumatoid arthritis	RA
Synovial fluid	SF
T regulatory cells	Treg
Toll-like receptors	TLR
Transforming growth factor beta	TGF-β
Tumor necrosis factor alpha	TNF-α
Vascular adhesion molecule 1	VCAM1

**Table 2 pharmaceuticals-15-00941-t002:** Phenotype of mesenchymal stem cells.

Positive ≥ 95%	Negative ≤ 2%
CD105	CD34
CD73	CD14, CD11b
CD90	CD79 alpha, CD19
	HLA-DR

**Table 3 pharmaceuticals-15-00941-t003:** The surface markers and identity of human mesenchymal stem cells.

Markers	Human MCSs	Properties/Functions
CD105/Endoglin	Bone marrow MSCs (BM MSCs), adipose tissue MSCs (ADSC), umbilical blood cord MSCs (UCB MSCs)	A type I transmembrane protein reported to induce activation and proliferation of endothelial cells and co-receptor for (transforming growth factor beta) TGF-β [34]
CD90/Thy-1	BM MSCs, ADSCs, UCB MSCs	Surface marker hypothesized to function in cell-cell and cell-matrix interactions, nerve regeneration, apoptosis, inflammation [32]
CD73/Ecto-5′-nucleotidase	BM MSCs, ADSC, UCB MSCs	Catalyzes the conversion at neutral pH of purine 5-prime mononucleotides to nucleosides, the preferred substrate being adenosine 5′-monophosphate (AMP) [32]
Stro-1	BM MSCs, Amnion MSCs (AMSCs), Synovial membrane derived MSCs	Cell surface antigen in human bone marrow cells capable of differentiating stromal cells with a vascular smooth muscle-like phenotype, adipocytes, osteoblasts and chondrocytes [35]
CD271/LNGFR/Low-affinity nerve growth factor receptor	BM MSCs, ADSC, Placenta MSCs, Wharton Jely derived MSCs (WJ MSCs), AMSCs, Chorion MSCs (CMSCs)	The specific markers for the purification of human BM-MSCs [36]
Oct-4/Octamer-binding protein 4	AMSCs, CMSCs	Transcription factors for pluripotency and self-renewal [35]
SSEA-4/Stage-specific embryonic antigen-4	BM MSCs, Synovial membrane derived MSCs	Stage-specific embryonic antigen and MSCs from whole human bone marrow [35,41]
CD146/MCAM/Melanoma cell adhesion molecule	BM MSCs, Synovial membrane derived MSCs, Pericytes	Receptor for laminin alpha 4, a matrix molecule that is broadly expressed within the vascular wall [37]
Sox11/SRY-Box Transcription Factor 11	BM MSCs	Marker downregulated during culture. Knockdown affect proliferation and osteogenesis potential [41]
CD349/Frizzled-9	AMSCs, CMSCs	A novel marker for isolation of MSC from placenta. Members of the ‘frizzled’ gene family encode 7-transmembrane domain proteins that are receptors for Wnt signaling proteins [35]

## Data Availability

Not applicable.

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
