# Peer review of "Interaction between Mesenchymal Stem Cells and the Immune System in Rheumatoid Arthritis"

_pharmaceuticals, 2022, doi:10.3390/ph15080941_

Round 1
Reviewer 1 Report
In this work, Bačenková and colleagues reviewed the interactions between MSCs and the immune system and its influence on the progression of RA. I believe the manuscript is well written, the information organized in a simple yet direct fashion and enough details about the reviewed works were provided. I recommend this manuscript to be returned to the authors with minor revisions, after which the manuscript can be deemed suitable for publication:
1. Please re-check the text to avoid any misinformation conveyed by grammatical errors. For example:
1.1. Line 218: “Chondo line” could be replaced by “Chondrogenic line” for better understanding;
1.2. Line 220: Please explain how can Interferon gamma have both “immunosuppressive” and “pro-inflammatory properties” at a site of inflammation.
Author Response
07/21/2022
Dear Sir,
thank you very much for yor comments to the topic. Following your instruction, entire text was checked throughly and corrected accordingly. Attached find a corrected version of the rewiew.
Loking forward to your reply.
Kind regards,
Sirecerely.
Darina Bačenková
Answer: Following your instructions, entire text was cheked throughly.
- Answer: I replaced Chondro line wih term chondrogenic line as recommended.
- Line 271: The autors added the text regarding the queation.

Reviewer 2 Report
In this manuscript, the authors reviewed the Interaction between mesenchymal stem cells and the immune system in rheumatoid arthritis. In my opinion, some issues should be further addressed and I hope the following comments could be helpful for improving their paper.
- Kindly add a clear introduction heading and explain, the background about mesenchymal stem cells and the immune system is little, the authors should enrich this part and emphasize the necessity of these cells for rheumatoid arthritis
- Authors focused on rheumatoid arthritis, but what are the distinguished properties and specific problems of rheumatoid arthritis therapy? The authors never discussed it.
- According to the applications, most, if not all are applicable for other kinds of diseases. Then why did the authors not expand the topic to other biomedical applications?
- Good quality figures are very important for a good review paper, kindly improve the quality of the figures
- The authors should summarize the current approaches to Interaction between mesenchymal stem cells and the immune system and compare their advantages and disadvantages in order to draw the reader's attention.
- This manuscript is well organized but lacks specific comparative analysis. What are the advantages of "Interaction between mesenchymal stem cells and the immune system "?
- Please revisit the entire manuscript for minor grammar issues.
- section 2 should also summarize in clear table form with examples.
- Authors also need to add an immune system heading and explain it and correlate it with RA.
- Future perspective and challenges is missing, kindly add it.
Author Response
07/21/2022
Dear Sir,
thank you very much for your comments to the topic. Following your instructions, entire text was checked throughly and corrected accordingly. Attached please find a corrected version of the review.
Looking forward to your reply.
Kind regards,
Darina Bačenková

Round 2
Reviewer 2 Report
Accepted in present form